# High and Low-Temperature Performance Evaluation and Microanalysis of SMCSBS Compound-Modified Asphalt

**DOI:** 10.3390/ma14040771

**Published:** 2021-02-06

**Authors:** Yu Sun, Dongpo He

**Affiliations:** 1School of Civil Engineering, Northeast Forestry University, Harbin 150040, China; donglin_2016@nefu.edu.cn; 2Department of Municipal and Environmental Engineering, Heilongjiang Institute of Construction Technology, Harbin 150000, China

**Keywords:** SMSSBS compound-modified asphalt, viscosity-temperature characteristics, high and low-temperature performance, microscopic features

## Abstract

The mixture of styreneic methyl copolymers (SMCs) normal temperature-modified asphalt and styrene-butadiene styrene block copolymer (SBS)-modified asphalt (SMCSBS) compound-modified asphalt was investigated in this study. The viscosity and temperature properties of compound modified asphalt (SMCSBS) were studied by Brookfield rotary viscosity test. Dynamic shear rheometer (DSR) and bending beam rheometer (BBR) were used to test SMCSBS compound modified asphalt with different SMC additions. Finally, the microstructure and physicochemical properties of SMCSBS were evaluated by scanning electron microscopy (SEM) and Fourier transform infrared spectroscopy (FTIR), and the modification mechanism of the SMCSBS was studied. The results show that the viscosity of the compound-modified asphalt added with SMC is improved, which is conducive to improving its workability. With the increase of SMC content, the high-temperature performance of the compound modified asphalt firstly increases and then decreases with the increase of SMC content. When the content of SMC is 12%, its high-temperature performance is the best. Compared with SBS-modified asphalt, the SMCSBS has better low-temperature performance, and the creep stiffness S and creep rate m of the SMC with different content are better than that of SBS. Finally, the microcosmic characteristics show that the SMC can give full play to its characteristics and can be uniformly dispersed in SBS modified asphalt. SMC is essentially a surfactant, which can reduce the viscosity and construction temperature by changing the surface tension and surface free energy of asphalt molecules. The curing agent of epoxy resin is slowly cross-linked and cured after contacting with air to form a certain strength, thus improving the road performance of the asphalt mixture.

## 1. Introduction

The start of warm mix asphalt technology is attributed to Professor Ladis Csanyi from Iowa State University in the United States, who first invented the preparation method of foamed asphalt in 1956, by pressing steam into the asphalt to form foamed asphalt [1]. In 1968, Mobil Oil Company in Australia obtained the patent of Professor Csanyi’s foamed asphalt preparation technology and registered the patent in 1971 [2]. Since then, countries such as Australia, Germany, France, Norway, and Japan have joined other development projects [3,4]. Until 1995, warm mix asphalt (WMA) was jointly developed by Shell and Kolo-veidekke in Europe and conducted field trials in the following year. In 1998, in order to reduce the production cost, the two companies started to use foamed asphalt and soft asphalt to produce warm mix asphalt. The prepared warm-mix asphalt mixture has a good performance after one year of comparative testing. In 2000, at the European International Asphalt Conference, two companies formally proposed the warm mix asphalt mixture, realizing the large-scale publicity and introduction of WMA. Subsequently, countries around the world began to study and apply WMA [5,6]. Since 2002, the United States has carried out the research and application of warm-mix asphalt mixture technology, paving their first warm-mix asphalt mixture pavement in 2004 [7]. Up to now, researchers such as Carmen Rubio M, Jamshidi Ali [8,9,10,11,12,13,14,15], and Chinese researchers such as Zhang Qisen, Shen Aiqin, Huang Xiaoming, Tan Yiqiu, etc. [16,17,18,19,20] have carried out fruitful research work on warm mix asphalt.

Styreneic methyl copolymers (SMC) is a kind of normal temperature asphalt modifier, with methyl styrene block copolymer extracted from waste plastics, waste rubber, and other methyl styrene polymer materials as the main raw material, and epoxy resin, epoxy resin curing agent, and other additives in a certain proportion with the polymer solution. After epoxy modification of the SMC, it has good compatibility with the asphalt material, and then the asphalt material is liquid at normal temperature and can be melted or dispersed in the asphalt to change the construction and workability of asphalt binder at normal temperature so that the asphalt and its mixture still have a certain fluidity at normal temperature and below zero [21]. At the same time, SMC normal temperature asphalt modifier has the function of first reducing the viscosity of asphalt and then enhancing the strength of asphalt, while the curing agent of epoxy resin is slowly cross-linked and cured after contacting with air to form a certain strength, thus improving the road performance of asphalt mixture [22]. In view of the characteristics of the SMC normal temperature asphalt modifier, this paper mainly evaluates its high and low-temperature performance during mixing and construction and analyzes its modification mechanism by combining it with microscopic properties.

At present, the SMC is used in many high and low-grade highways in China. Sun Zhigang [23] summarized the quality control technology and tracking observation of SMC normal temperature modified asphalt mixture in road maintenance engineering in the Qingyang area. Zhao Lina [24] added SMC normal temperature modified asphalt into the base asphalt binder to improve road performance. Qu Hongwei [25] proposed to apply SMC normal temperature modified asphalt in highway covering construction and elaborated the specific construction points, pointing out that SMC normal temperature modified asphalt has the advantages of energy saving and environmental protection, convenient construction, and significant economic benefits. Zhu Jianfeng et al. [26] dried the asphalt containing a certain amount of organic content in an oven at 100 °C and took it out every hour to measure its mass loss and standard viscosity. The results show that the organic solvent content, compaction function, curing temperature, and cycle affect the formation of strength to a certain extent, and the gradation is an important factor for the formation of strength; with the volatilization of organic solvents, the strength of SMC normal temperature asphalt mixture gradually increases. When the organic solvents are all volatilized, the strength is finally formed. Luo Haoyuan et al. [27] used infrared spectroscopy to test mixing temperature, volume index stability, residual stability, freeze-thaw splitting strength ratio, bending strain failure strain of the mixture under different SMC content, and the results showed that the appropriate SMC content was 6–10% under SMC-13 (the SMC modified asphalt mixture is mixed with asphalt concrete (AC-13)) gradation and 6–8% under SMC-20 (the SMC modified asphalt mixture is mixed with AC-20) gradation. Some scholars also studied the performance of SMC recycled asphalt mixture at normal temperature with high Reclaimed Asphalt Pavement (RAP) content [28]. The application of an SMC normal temperature modifier can ensure the road performance at normal temperature and increase the amount of RAP. Wang Jianjun [29] used SBS and SMC to prepare the compound modified constant temperature warm-mix asphalt and applied it to AC-13 (the mixture ratio of asphalt mixture is asphalt concrete (AC-13)) asphalt mixture. Through laboratory tests, the high-temperature performance, low-temperature performance, and moisture susceptibility of the mixture were evaluated [30], and performance analysis was conducted for road material.

Above all, on the basis of existing research, this paper adopts the SBS and SMC to prepare compound-modified normal temperature mixed asphalt, analyzes the influence of different content of modifier on the asphalt performance, the Brinell rotary viscosity test, dynamic shear rheometer (DSR), and bending beam rheometer (BBR) test method, and discusses the performance of SMCSBS compound-modified asphalt in order to test results to promote the application of normal temperature compound-modified asphalt to provide data reference.

## 2. Materials and Methods

### 2.1. Materials

The base asphalt binder used in the test is Panjin 90# asphalt from Panjin, China. Styrene-butadiene styrene block copolymer (SBS) is an SBS modifier produced by Sinopec Asset Management Co., Ltd., Yueyang, Hunan, China. Styreneic methyl copolymers (SMCs) are provided by Ningxia Ruitai Tiancheng New Material Technology Co., Ltd., Yinchuan, China. In the SMCSBS, the mass fraction of SBS is 4%, and the SMC content is 8%, 10%, 12%, and 14%, respectively. The technical performance indexes of 90# base asphalt binder, SBS-modified asphalt, and SMC are shown in Table 1, Table 2 and Table 3, respectively.

### 2.2. Methods

In the experiment, SMCSBS compound modified asphalt was prepared by using a high-speed shear mixing emulsifier. The base asphalt was heated to the molten state, and then, according to the processing temperature requirements of modified asphalt in JTG F40-2004, SBS and SMC were used for high-speed shear at about 175 °C, the rotation speed was 5000 r/min, the shearing time was 60 min, and after shearing, mixing 40 min to uniformly prepare SMCSBS compound-modified asphalt. Under the condition of the same amount of SBS, different amounts of SMC normal temperature modified asphalt were added, and then the penetration test, ductility test, softening point test, DSR test, and BBR test were carried out on the SMCSBS compound-modified asphalt with different SBS content.

The compound-modified asphalt samples with SMC ratios of 8%, 10%, 12%, and 14% were prepared respectively for penetration, ductility test (JTG E20-2011), and softening point test (JTG E20-2011) [31]. The temperature of the penetration test was 25 °C, the load was 100 g, and the time was 5 s. The ductility test temperature was 5 °C, and the tensile rate was 5 cm/min. Two parallel tests were carried out on the SMCSBS with different content.

Brookfield rotary viscosity test (AASHTO T312, Washington, DC, USA), to determine the rotational viscosity of the SMCSBS, the mixing temperature must be at the viscosity 0.17 ± 0.02 Pa·s, the compaction temperature must be at the viscosity 0.28 ± 0.02 Pa·s. Two parallel tests were carried out on the SMCSBS with different content.

The sensitivity of the SMCSBS to temperature was evaluated by bending beam rheometer (BBR, Washington, DC, USA) and dynamic shear rheometer (DSR, Washington, DC, USA) to determine the low-temperature evaluation index and high-temperature rheological properties of the compound modified asphalt [32].

For the dynamic shear rheology (DSR) test, the stress control mode is adopted in this research. The stress level is maintained at 0.1 kPa and the angular frequency is 10 rad/s. The diameter of the specimen was 8 mm (thickness was 2 mm) (AASHTO)T315-09). Two parallel tests were carried out on the SMCSBS with different content.

Through the bending beam rheometer (BBR) test, the creep stiffness modulus S and creep rate m of asphalt were obtained, and the low temperature crack resistance index m/S was used to evaluate the low-temperature crack resistance of asphalt. The specimen size is length, width, and thickness (102 mm × 12.7 mm × 6.35 mm) (AASHTO T313-09); two parallel tests were carried out on the SMCSBS with different content.

By observing the microstructure changes of the asphalt binder, studying the interaction between the polymer and the asphalt binder, scanning electron microscopy (SEM) and Fourier transform infrared spectroscopy (FTIR, Beijing, China) can observe the asphaltene aggregation and physicochemical interaction [33].

In this experiment, using the scanning electron microscope (SEM, Hitachi TM3030, Tokyo, Japan) according to JB/T 6842–1993 (a Chinese Industry Standard for the test methods of electron microscopy), the SEM test of SBS and SMC compound modified asphalt with different content was carried out.

The Fourier transform infrared spectroscopy (FTIR) is to irradiate a beam of infrared rays of different wavelengths onto the molecule of a substance, and certain infrared rays of specific wavelengths are absorbed to form the infrared absorption spectrum of this molecule. Each molecule has a unique infrared absorption spectrum determined by its composition and structure, which allows structural analysis and identification of the molecules. In this study, an infrared spectrometer was used to obtain the infrared spectra of SBS- and SMC-modified asphalt with different content by attenuating the total reflection method. The detection wavelength ranged from 4000 cm^−1^ to 500 cm^−1^, and the peaks of functional groups in the spectrum were analyzed.

## 3. Results

### 3.1. Basic Technical Indexes of SMCSBS with Different SMC Content

Referring to the technical standard of the liquid petroleum asphalt testing method, the main technical indexes of normal temperature modified asphalt with different SMC content were obtained through testing. The test results are shown in Table 4.

It can be seen from Table 4, with the increase of SMC content, the penetration degree of SMCSBS at 25 °C increases, the softening point decreases, and the ductility increases at 5 °C. When the SMC content increases to 10%, it reaches the maximum ductility, indicating that the compound-modified asphalt greatly increases the ductility of the binder. The results show that the addition of the SMC can improve the high and low-temperature performance of SBS.

### 3.2. Rotational Viscosity

#### 3.2.1. Mixing Temperature and Compaction Temperature

The mixing temperature and compaction temperature are determined by the chart, so the rotational viscosity of the mixing temperature must be 0.17 ± 0.02 Pa·s, and the rotational viscosity of the compaction temperature must be 0.28 ± 0.02 Pa·s [34]. Table 5 and Figure 1 show the results of the rotational viscosity tests obtained by mixing (M) and compacting (C) temperatures. The mixing temperature and compaction temperature decreased with the addition of SMC, and the decreasing rate tended to be gentle with the increase of the content. The mixing temperature of SMCSBS compound-modified asphalt is about 30 °C lower than that of SBS asphalt, and the temperature dropped to between 100 °C and 120 °C.

As can be seen from Table 5, in terms of reducing the mixing temperature and the compaction temperature, the content of SMC is improved and lower than that of modified asphalt, indicating that the workability of compound material is better. In addition, the reduced temperature brings energy efficiency to road production. Thives and Guisi [35] believe that the use of warm-mix technology can significantly reduce energy consumption and CO_2_ emissions compared with hot-mix asphalt production.

#### 3.2.2. Viscous-Temperature Characteristics

Figure 2 shows the change of the viscosity with the temperature of the compound-modified asphalt with different content. The viscosity of the compound modified asphalt is compared with that of SBS, especially the SMC (8%) increases. The viscosity of 14% in SMC with different content was lower.

As shown in Figure 2, the log-log of the viscosity is linearly related to the log of temperature. The linear relationship can be described by the Saal model, which is shown in Equation (1) as
log log(η) = *n* − *m* log (*T* + 273.2),(1)
where η is Brookfield viscosity, mPa·s; *T* is temperature, °C; and *m* and *n* are linear regression coefficients.

The value of m indicates the sensitivity of viscosity to temperature. The greater the absolute value of *m*, the higher the sensitivity of viscosity to temperature. The fitting *m* values of the Saal model for different specimens are listed in Table 6. The addition of SMC reduces the sensitivity of binder viscosity to temperature. The viscosity of 8% SMCSBS has the lowest temperature sensitivity, while the viscosity of 12% SMCSBS is more sensitive to temperature changes, which is beneficial to improve its workability.

### 3.3. High-Temperature Performance Analysis

#### 3.3.1. Performance Grade (PG)

DSR dynamic shear test and BBR bending beam rheometer test were carried out on compound modified asphalt (SMCSBS) with 8%, 10%, 12%, and 14% content [36]. According to the Superpave specification, PGs are obtained, |*G**|/sin(δ) is higher than 1.00 kPa [37]. The test results are shown in Table 7.

#### 3.3.2. Analysis of the Rut Factor (*G**/sinδ)

In Superpave, the ratio of *G** and δ to the rutting deformation resistance of asphalt pavement is used as the evaluation index of the rutting deformation resistance of asphalt materials, and it is called as the factor (*G**/sin δ). The larger the coefficient, the stronger the asphalt’s resistance to deformation and the better its high-temperature performance. By using a dynamic shear rheology (DSR) test, the *G**/sin δ results at different temperatures and different SMC content were analyzed. It can be seen from Figure 3 that (1) *G**/sin δ decreases with the increase of temperature; with the increase of SMC content, the high-temperature performance of the composite-modified asphalt increases first and then decreases; (2) the maximum factor (*G**/sin δ) is when the SMC content is 12%; and (3) when the SMC content is 8% and 10%, the *G**/sin δ curves almost coincide, and the high-temperature performance is very close.

It can be seen that the high-temperature performance of SMC on the compound-modified asphalt improves at first and then decreases with the increase of the content. At the same time, the study [27] also proposed that the softening point would decrease when the SMC content was greater than 12%, suggesting that the SMC content should not exceed 12%.

#### 3.3.3. Critical Temperature

In order to accurately describe the high-temperature deformation resistance of SMCSBS asphalt with different content, the critical temperature was calculated through linear fitting analysis, that is, the linear regression analysis was carried out on the curve, and the corresponding temperature at (*G**/sinδ) = 1 kPa was obtained. The higher the critical temperature, the better the high-temperature performance of asphalt material [38].

According to the corresponding factor (*G**/sinδ) of *T* at different temperatures, the curve of (*G**/sinδ)-T was established. To facilitate calculation, the logarithm of the factor was taken as the dependent variable to establish the logarithm curve of the factor lg(*G**/sinδ)-*T*, as shown in Figure 4.

According to the regression fitting linear equation, the critical temperature corresponding to (*G**/sinδ) = 1 kPa can be obtained, that is, lg (*G**/sinδ) = 0. The results are shown in Table 8.

In conclusion, we can see that the high-temperature performance of SMC compound asphalt with different content increases firstly with the increase of SMC content, and then the high-temperature performance decreases with the increase of content.

### 3.4. Low-Temperature Performance Analysis

#### 3.4.1. Performance Grade (PG)

BBR test was carried out for SMCSBS with different content. The test results are shown in Figure 5 and Figure 6. It can be seen from Table 3 that the compound asphalt mixed with SMC modifier can meet the demand at −34 °C, and its low-temperature performance is better than that of SBS asphalt. At the same temperature, the creep stiffness S of 8% SMC is the largest and decreases with the increase of the content of SMC. The creep velocity m of 14% SMC is the largest, while that of 12% SMC is the smallest.

#### 3.4.2. Creep Stiffness S and Creep Rate m

Through the rheological test of the curved beam, the creep stiffness modulus S and creep rate m can be obtained. These two indexes can well describe the low-temperature performance of asphalt. The smaller the S value, the better the low-temperature flexibility. The greater the m value, the better the stress relaxation and crack resistance of asphalt [39]. It can be seen from Figure 5 that, at the same temperature, the creep stiffness modulus S value of the SMC asphalt with different content is lower than that of the SBS. The creep rate m value of the SMC asphalt with different content is higher than that of the SBS. The results show that the low-temperature performance of the SMC asphalt with different content is better than that of the SBS.

In addition, the creep stiffness modulus S values of the four different compound asphalt are as follows: 8% > 10% > 12% > 14%, indicating that the higher the SMC content, the smaller the S value; the creep rate m generally changes as follows: the higher the SMC content is, the higher the m value is, however, with the decrease of temperature, the performance of SMCSBS with 12% content changes most obviously. From −30 °C to −18 °C, with the decrease of temperature, the creep stiffness modulus of four kinds of asphalt with different content increases gradually, while the creep rate of m decreases gradually. This indicates that the low-temperature flexibility, stress relaxation ability, and low-temperature crack resistance of four kinds of SMC compound modified asphalt and SBS with different content decrease with the decrease of temperature.

#### 3.4.3. Crack Resistance Index m/S at Low Temperature

In order to better evaluate the low-temperature crack resistance of asphalt, the ratio m/S of creep stiffness S to creep rate m is used for comparative analysis. The larger the m/S is, the better the low temperature cracking resistance of asphalt is, and vice versa. Figure 6 shows the change curve of m/S value of SBS and SMCSBS asphalt with different content with temperature. It can be seen that the SMCSBS has better low temperature cracking resistance than the SBS asphalt. Moreover, in the range of −18 °C to −30 °C, the m/S value of the compound-modified asphalt decreases with the decrease of temperature, indicating that the low-temperature crack resistance of compound modified asphalt decreases with the decrease of temperature. However, at different temperatures, the value of 14% m/S is the maximum, and the value of 8% m/S is the minimum, indicating that the higher the SMC content, the better the low-temperature crack resistance. In the temperature range of −18 °C, the m/S value of the compound modified asphalt with different content varies greatly; with the decrease of temperature, the m/S values of the compound modified asphalt with different content were close to each other. When the temperature was lower than −24 °C, the m/S values of 10% and 12% were infinitely close. At −30 °C, their values were almost the same, indicating that there was little difference in their low-temperature performance with the decrease of temperature. Obviously, the SMCSBS has better low-temperature performance compared with the SBS. However, with the increase of SMC content, the high-temperature performance gradually decreased. Therefore, the high and low-temperature performance of SMCSBS should be studied in combination.

### 3.5. Micro Analysis

#### 3.5.1. SEM

The distribution of SBS and SMC modifiers in SBS asphalt was observed by scanning electron microscopy. The test results are shown in Figure 7a–l.

The SMC can give full play to its characteristics and can be evenly dispersed in the SBS asphalt, and the whole compound-modified asphalt has its own characteristics so that the compound modified asphalt can obtain better performance.

SEM can be used to observe the distribution of SMC and SBS in the base asphalt and the micromorphology of the compound-modified asphalt. The results are shown in Figure 7a–l. It can be seen from Figure 7d–l, after sufficient mixing, SMC can be uniformly dispersed in the asphalt and form a certain spatial network structure with SBS asphalt, thus improving the stability of the compound modified asphalt system.

#### 3.5.2. FITR

In order to understand the modification mechanism of the SMC, this study used FTIR to analyze the SBS, 8% SMCSBS, 10% SMCSBS, and 12% SMCSBS asphalt. The samples were taken 3 h after the preparation and were in a relatively stable state for analysis. The results are shown in Figure 8. By comparing the change of peak position and peak height, the following phenomena can be observed:

Compared with the SBS, there is an N-H bond (secondary amine) stretching vibration peak at 2390 cm^−1^; the position at 1035 cm^−1^ belongs to the characteristic range of $\nu_{c-o-c}^{s}$. After the SBS asphalt was added to the SMC, the peak value of 966 cm^−1^ was still obvious. In addition, the characteristic peak of bisphenol A type epoxy resin (
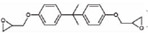
) exists in SMC [40]. The characteristic peak at 742–1035 cm^−1^ proves that there are a few end-group epoxy rings in the SMC, but the characteristic peaks such as the benzene ring basically disappear. This phenomenon should be caused by the volatilization of bisphenol A epoxy resin as a solvent.

### 3.6. Study on Modification Mechanism of the SMCSBS

Through the above analysis and reference literature [40,41], it can be seen that the characteristic change of the SMC is a basic reaction of the action principle of cationic surfactant, and it can play a lubrication role by changing the surface tension of the solution. In addition, after the evaporation of epoxy resin, its characteristics are similar to the SBS asphalt. To conclude, the SMC is essentially a surfactant, which can reduce its viscosity and construction temperature by changing the surface tension and surface free energy of asphalt molecules. The curing agent of epoxy resin is slowly cross-linked and cured after contacting with air to form a certain strength, thus improving the road performance of the asphalt mixture.

## 4. Conclusions

Based on the obtained results, the following conclusions can be drawn:The results show that the viscosity of the compound asphalt with SMC is sensitive to temperature change—8% SMCSBS viscosity has the lowest temperature sensitivity, while 12% SMCSBS viscosity is more sensitive to temperature changes, which is conducive to improving its workability. At the same time, careful handling should be paid attention to during construction;High-temperature performance of SMCSBS compound modified asphalt—the high-temperature performance of SMC compound asphalt with different content increases firstly with the increase of SMC content, and then the high-temperature performance decreases with the increase of SMC content. When the SMC content is 12%, the factor (*G**/sinδ) was the highest;Low-temperature performance of SMCSBS compound modified asphalt—compared with SBS, the SMCSBS has a better low-temperature performance. The low-temperature performance of SMCSBS increases with the increase of SMC content, and the creep stiffness modulus S value of four kinds of composite asphalt with different content is 8% > 10% > 12% > 14%. The creep rate m value showed that the performance of SMCSBS with 12% content changed most obviously. However, with the increase of SMC content, the high-temperature performance gradually decreased. Therefore, the high and low-temperature performance of SMCSBS should be studied in combination;In addition, the PG grades with different SMC content of 8%, 10%, 12%, and 14% are PG58-26, PG64-32, PG64-32, PG58-32, respectively, indicating that the SMCSBS has good low temperature and can meet the use requirements under poor low temperature and harsh conditions;In this paper, by means of microscopic analysis, SEM and FTIR results show that the SMC can give full play to its characteristics and can be evenly dispersed in the SBS asphalt, SMC is essentially a kind of surface active agent, by changing the surface tension and freeing the surface energy of asphalt molecules to achieve the purpose of reducing its viscosity and temperature; it also contains the curing agent of epoxy resin, crosslinking curing agent slowly after it is exposed to air to form a certain intensity.

## Figures and Tables

**Figure 1 materials-14-00771-f001:**
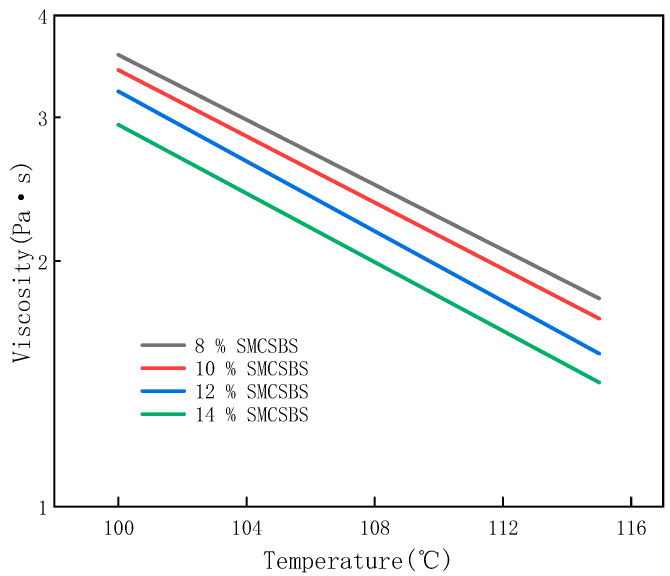
Viscosity-temperature curve of SMC normal temperature-modified asphalt and SBS-modified asphalt (SMCSBS) with different content.

**Figure 2 materials-14-00771-f002:**
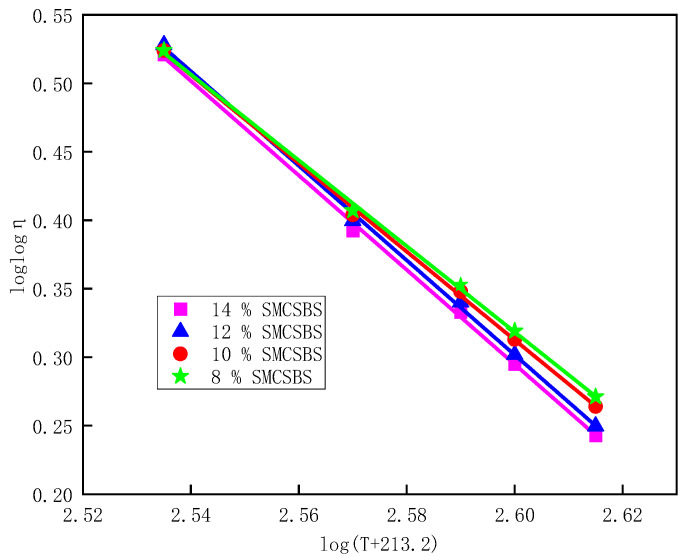
Temperature dependence of SMCSBS with different content.

**Figure 3 materials-14-00771-f003:**
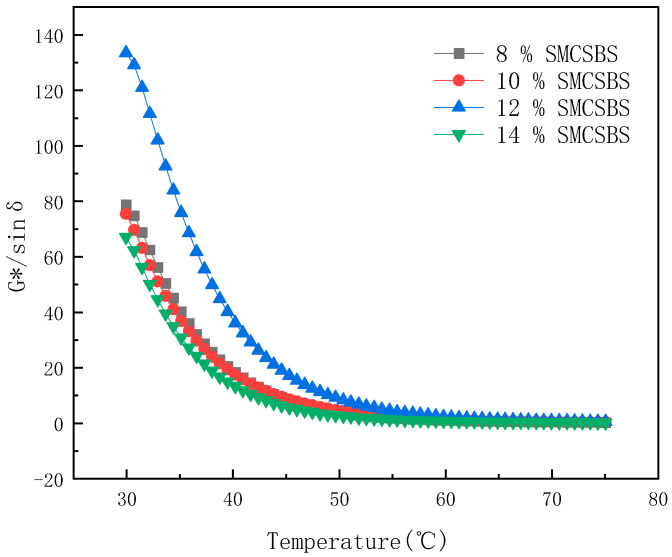
Variation curve of *G**/sin δ of different SMCSBS content with temperature.

**Figure 4 materials-14-00771-f004:**
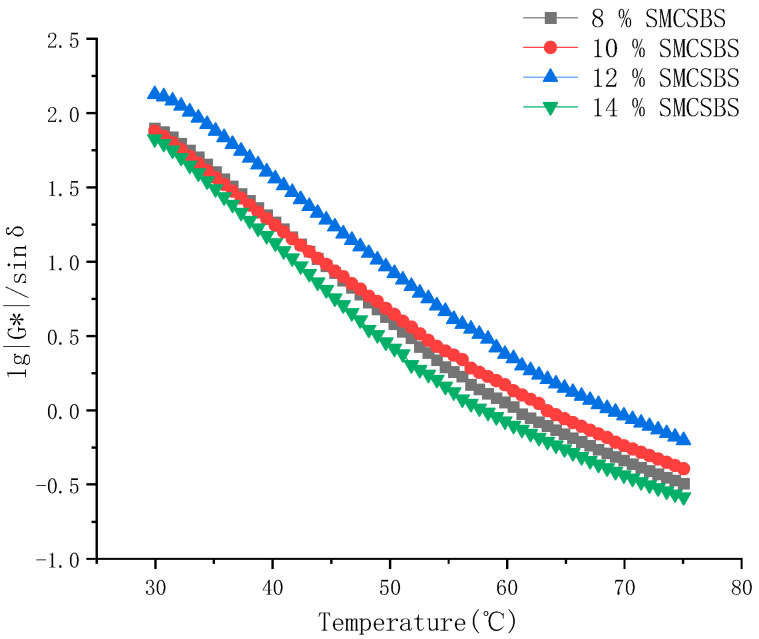
lg (*G**/sin δ)-T curve of with different SMCSBS content.

**Figure 5 materials-14-00771-f005:**
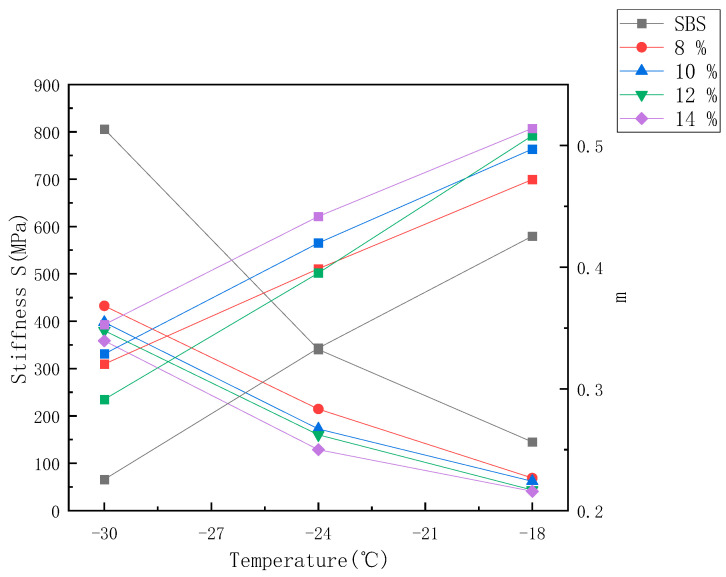
Curves of creep stiffness S and creep rate m of SBS and SMCSBS with different content with temperature.

**Figure 6 materials-14-00771-f006:**
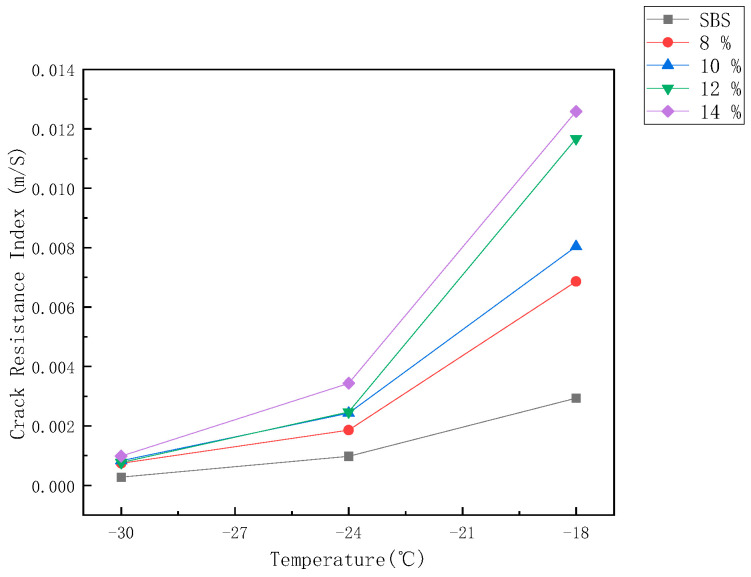
Change curve of m/S value of SBS and SMCSBS with different content with temperature.

**Figure 7 materials-14-00771-f007:**
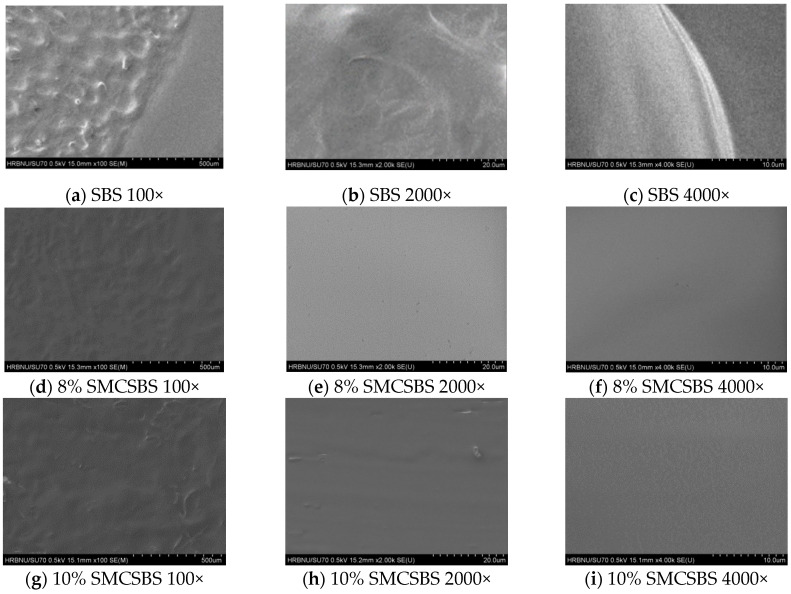
(**a**) SBS 100×; (**b**) SBS 2000×; (**c**) SBS 4000×; (**d**) 8% SMCSBS 100×; (**e**) 8% SMCSBS 2000×; (**f**) 8% SMCSBS 4000×; (**g**) 10% SMCSBS 100×; (**h**) 10% SMCSBS 2000×; (**i**) 10% SMCSBS 4000×; (**j**) 12% SMCSBS 100×; (**k**) 12% SMCSBS 2000×; and (**l**) 12% SMCSBS 4000×.

**Figure 8 materials-14-00771-f008:**
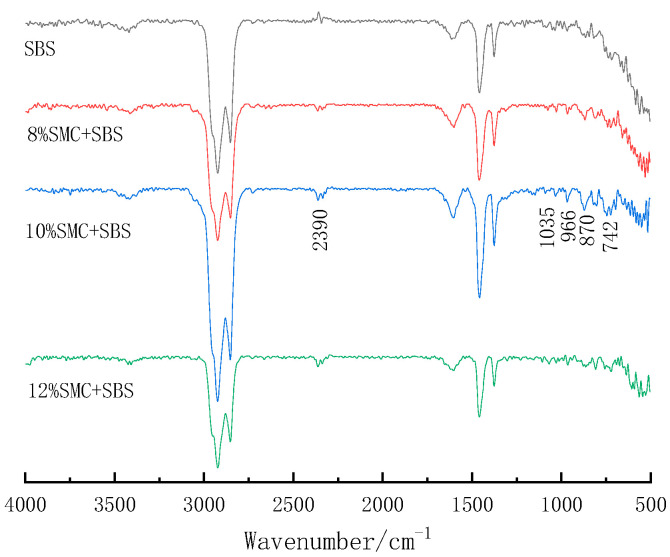
Infrared spectra of SBS and SMCSBS with different content.

**Table 1 materials-14-00771-t001:** Main technical indexes of 90# base asphalt binder.

Indexes	Unit	Results	Technical Indicators ^1^
Penetration (25 °C)	0.1 mm	84	80~100
Softening Point	°C	49	44
Ductility (15 °C)	cm	>100	100

^1^ The technical requirements of the test shall be in accordance with China’s current Technical Code for Construction of Highway Asphalt Pavement (FTG F40-2004).

**Table 2 materials-14-00771-t002:** Main technical indexes of styrene-butadiene styrene block copolymer (SBS) modified asphalt.

Indexes	Unit	Results	Technical Indicators ^2^
Penetration (25 °C)	0.1 mm	90	80~100
Softening Point	°C	50	45
Ductility (15 °C)	cm	>100	100

^2^ The technical requirements of the test shall be in accordance with China’s current Technical Code for Construction of Highway Asphalt Pavement (FTG F40-2004).

**Table 3 materials-14-00771-t003:** Technical indexes of styreneic methyl copolymers (SMCs) modifier.

Indexes	Unit	Results	Technical Indicators ^3^
Appearance	-	A brown viscous liquid	-
Density	g/cm^3^	0.92	0.8~1.0
Flash point, not less than	°C	92	170
Rubber hydrocarbon content, not less than	%	93	85
Volatile organic benzene content, not more than	%	0.01	0.1

^3^ The test technical requirements shall be implemented in accordance with China’s local standard “Technical Guide for Warm-mixed modified asphalt Consumption Layer” (2018-04-09).

**Table 4 materials-14-00771-t004:** Studies the process indexes of modified asphalt with different content.

SMC Content/%	25 °C Penetration/0.1 mm		Softening Point °C		5 °C Ductility		35 °C Viscosity/Pa·s
	Average	Standard Deviation	Average	Standard Deviation	Average	Standard Deviation	
8	191	0.211	74	0.050	89.7	0.531	1662.34
10	228	0.325	56.5	0.400	>100	0	671.29
12	248	0.561	55	0.636	>100	0	373.73

**Table 5 materials-14-00771-t005:** Temperature changes (mixing and compaction).

Additive Content (%)	Temperatures (°C)	
	Mixing Temperature (M)	Compaction Temperature (°C)
SBS	165–170	150–155
8% SMCSBS	120.27–118.26	114.75–112.74
10% SMCSBS	118.98–117.01	113.56–111.59
12% SMCSBS	116.72–114.86	111.60–109.73
14% SMCSBS	115.11–113.21	109.89–107.99

**Table 6 materials-14-00771-t006:** The Saal model parameters fitted with different content.

Specimens	Absolute Value of *m*	*n*
8% SMCSBS	3.13394	8.46664
10% SMCSBS	3.2235	8.69367
12% SMCSBS	3.45191	9.2766
14% SMCSBS	3.44914	9.2625

**Table 7 materials-14-00771-t007:** Performance grade (PG) results of SBS and the SMCSBS.

Type	Specimens	PG (°C)
Pure	SBS	64–28
SMC Compound Modified Asphalt	8% SMCSBS	58–34
	10% SMCSBS	64–34
	12% SMCSBS	64–34
	14% SMCSBS	58–34

**Table 8 materials-14-00771-t008:** The critical temperature of modified asphalt.

Contents	Relation	*R* ^2^	Standard Deviations	Variation Coefficient	Critical Temperature/°C
8% SMCSBS	*y* = −0.05586*x* + 3.48612	0.98456	0.00089	1.14673	62.41
10% SMCSBS	*y* = −0.05173*x* + 3.32187	0.98915	0.00069	1.36044	64.22
12% SMCSBS	*y* = −0.05485*x* + 3.74342	0.99165	0.00064	0.85154	68.25
14% SMCSBS	*y* = −0.05521*x* + 3.32741	0.97700	0.00108	1.74475	60.27

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
