# Peer review of "High and Low-Temperature Performance Evaluation and Microanalysis of SMCSBS Compound-Modified Asphalt"

_materials, 2021, doi:10.3390/ma14040771_

Round 1
Reviewer 1 Report
Congratulations to authors. Very interesting paper regarding the use of Styrenic Methyl Copolymers (SMC) in Polymer Modified Bitumen (PMB) in order to improve the mechanical performance of bituminous mixtures.
In general, the paper is well presented but some information shall be clarified or completed:
- Some sentences are very similar (although they are not exactly the same) with others included in the paper “High and Low Temperature Performance and Fatigue Properties of Silica Fume/SBS Compound Modified Asphalt" (by “Xuewen Zheng, Wenyuan Xu, Huimin Feng and Kai Cao) already published in Materials (MDPI). Please, try to rewrite some sentences;
- It seems appropriate to include a list of acronyms (or “Abbreviations”) before “References”;
- Under the International System of Units (SI) recommendations (2019), a space must separate the number and the percentage symbol (“## %” and not “##%”);
- The same recommendation is made for Celsius degree symbol (“## °C” and not “##°C”);
- Line 8: can you include any institutional e-mail?
- Line 9: define here the acronym “SMC”;
- Lines 32/34: the start of warm mix asphalt technology is attributed to Prof. Ladis Csanyi from Iowa State University (USA), who first produced a foamed bitumen in 1956. After that and until 1995, other developments were added in Australia and then in Germany. So, you have to complete this historical review (please consult other references);
- Line 62: what do you mean by "matrix asphalt”? (“mastic asphalt”? or “neat asphalt”? or “asphalt”?);
- Lines 77 & 82: please define “SMC-13”, “SMC-20” and “AC-13”;
- Line 83: where is written “water stability” do you want to write “water sensitivity”?
- Line 89: what do you mean with “SBS/SBS compound modified asphalt”?
- Line 94: again, what is “matrix asphalt (MA)”?
- Lines 98/99: “In the SMCSBS, the mass fraction of SBS is 4%, and the SBS content is 8%, 10%, 98 12% and 14%, respectively”, it’s correct?
- Tables 1 to 3: where can we find these “Technical indicators”?
- Line 116: “… penetration test was 5℃” or “25 ℃”?
- Line 127: “For dynamic shear rheology (DSR) test, stress control mode is adopted in this paper.” Do you want to write “In this paper” or “in this research”?
- Line 134: where is written “… 102×12.7×6.35 mm …” must be “… 102×12.7×6.35 mm3 …” or “… 102 mm × 12.7 mm × 6.35 mm …”;
- Line 141: what is the “JB/T 6842-1993”? Explain that this is a Chinese Industry Standard for the test method of SEM;
- Line 143: after you define “The Fourier transform infrared spectroscopy (FTIR)” then you can use only the acronym FTIR;
- Table 4 and line 158: “… penetration test was 5℃” or “25 ℃”?
- Table 5: you can improve two column titles: “Temperatures (℃) and then "Mixing (M)" / "Compaction (C)”;
- Figure 3 and Figure 4: one of these figures is expendable;
- Table 8: please include a new column with the variation coefficient (=standard deviation / mean);
- Lines 279/882: you can find a better way to present these indications;
- Figure 8: please remove all the hidden data (that appears when we place the cursor over these images);
- Again, Figure 8: you can reorganize the presentation of these images. For example, considering three columns (one for each magnification, 100× / 2000× / 4000×);
- Line 332: please correct “3.5.2. FITT” by “3.5.2. FTIR”;
- Figure 9: please correct “Warenumber/cm-1” by “Wavenumber/cm-1”;
- You can also include some information regarding one important aspect for the application industry: average cost of this polymer (in comparison with a conventional PMB, for example, SBS);
- Section 4. (Conclusions): some conclusions can be better summarized. Please also include (where appropriate) the relevant drawbacks or constraints;
- References: I advise the Authors to include some other recent references.
Author Response
Dear Professor:
Thank you very much for your comments. Based on your comment and request, we have made extensive modification on the original manuscript. Please see the attachment.
Yours,
Yu Sun

Reviewer 2 Report
- You use very colorful language, that is not always appropriate for research paper. Example: “Zhu Jianfeng et al. [22] baked the asphalt”. Word baked is not appropriate in this context. You should revise all manuscript and improve your language. There are also some typos.
- You have used ~25 references in the introduction, but it is still kind of limited.
- State clearly your objectives.
- How did you choose the amount of polymer?
- Why have you chosen 175C for mixing?
- Can you explain the behavior of the material in Figures 3 and 4? How many samples of each combination have you tested?
- Figure 4 and 5 contains the same data?
- Conclusion 1: How can you improve viscosity? Binder more sensitive to temperature change is harder to work and needs more careful handling.
- Please, improve your conclusions. It seams like you just repeat results.
Author Response

(The authors gave the same response as above.)

Round 2
Reviewer 2 Report
Thank you for addressing my comments.